# Pregnant Dutch Women Have Inadequate Iodine Status and Selenium Intake

**DOI:** 10.3390/nu14193936

**Published:** 2022-09-22

**Authors:** K. Clara Mayunga, Melany Lim-A-Po, Janniek Lubberts, Eline Stoutjesdijk, Daan J. Touw, Frits A. J. Muskiet, D. A. Janneke Dijck-Brouwer

**Affiliations:** 1Department of Laboratory Medicine, University Medical Center Groningen, University of Groningen, Hanzeplein 1, 9713 GZ Groningen, The Netherlands; 2Radboud Laboratory for Diagnostics, Radboud University Medical Center, Geert Grooteplein Zuid 10, 6525 GA Nijmegen, The Netherlands; 3Department of Clinical Pharmacy & Pharmacology, University Medical Center Groningen, University of Groningen, Hanzeplein 1, 9713 GZ Groningen, The Netherlands

**Keywords:** iodine, selenium, pregnancy, supplement, urine

## Abstract

Iodine and selenium are essential for thyroid hormone synthesis. Iodine and selenium interact. Pregnancy increases the maternal iodine requirement. We previously reported inadequate iodine status in pregnant Dutch women. Since little is known about their selenium intake, we investigated the iodine status and selenium intake in relation to iodine and selenium supplement use during pregnancy. Iodine status was established in 201 apparently healthy pregnant women as 24 h iodine excretion (24H-UIE; sufficient if median ≥225 µg), iodine concentration (24H-UIC; ≥150 µg/L) and iodine/creatinine ratio (24H-UICR; ≥150 µg/g). Selenium intake was calculated from 24 h selenium excretion. Iodine status in pregnancy proved insufficient (medians: 24H-UIE 185 µg; 24H-UIC 95 µg/L; 24H-UICR 141 µg/g). Only women taking 150 µg iodine/day were sufficient (median 24H-UIE 244 µg). Selenium intake was below the Estimated Average Requirement (EAR; 49 µg/day) in 53.8%, below the Recommended Dietary Allowance (RDA; 60 µg/day) in 77.4% and below the Adequate Intake (AI; 70 µg/day) in 88.7%. Combined inadequate iodine status and selenium intake <RDA was found in 61%. Women who want to become pregnant should, consistently with WHO and ETA recommendations, be advised to use a 150 µg iodine-containing supplement. Concomitant selenium supplementation should be added to this advice, at least in The Netherlands.

## 1. Introduction

Iodine and selenium are essential for thyroid hormone synthesis. Iodine deficiency may cause clinical or subclinical hypothyroidism. Pregnancy augments both maternal thyroid hormone synthesis and iodine needs with approximately 50%. The developing fetal brain needs iodine in the first stage of development. Mild or subclinical hypothyroidism in early pregnancy may adversely impact neurological development [1,2]. At later stages, the fetus starts synthesizing its own thyroid hormone from transplacentally transported iodine and becomes less dependent on a preformed maternal source. Iodine insufficiency during pregnancy is associated with an increased risk of obstetrical complications [2], suboptimal cognitive development [3] and educational outcomes [4] and attention deficit hyperactivity disorder [5] of the offspring. A recent study showed a curvilinear association between the urinary iodine/creatinine ratio (UICR) in pregnancy and the offspring’s brain gray matter volume at 10 years [6]. Early gestation iodine supplementation of pregnant women with mild hypothyroidism improved offspring neurological development, as compared to later or no supplementation [7].

The sodium–iodine symporter (NIS) is present not only in the thyroid, mammary glands and the placenta but also in salivary glands, stomach, rectum, kidneys [8,9], (fetal) thymus [10] and other organs [11]. Approximately 50–70% of the total body iodine pool is located in extrathyroidal tissues, which together with the wide organ distribution of the NIS points at other important iodine functions, such as its use as an antioxidant, in the immune system and as an antimicrobial agent [12].

The thyroid contains the most selenium per gram of all organs [13]. Its local function is in thyroid hormone synthesis and iodide liberation, catalyzed by the selenium-containing deiodinases (DIOs), and also in the destruction of locally synthesized redundant hydrogen peroxide by seleno-enzymes glutathione peroxidase and thioredoxin reductase [14,15].

Both low and high iodine status is associated with thyroid autoimmune disease [16]. The common denominator might be low selenium status, prompting Rayman to conclude that regions with deficient iodine intake, more-than-adequate iodine intake and high iodine intake may need more selenium, owing to the capacity of selenoproteins to protect the thyroid from excessive hydrogen peroxide and inflammation [17]. Selenium and selenoproteins are also involved in the immune system and brain functioning [18]. In the perinatal period, selenium plays a role in fertility, offspring immunity and neurological development [19], pregnancy-induced hypertension, preeclampsia, preterm birth and miscarriage [20,21].

Seaweed, and seafood in general, are rich sources of both iodine and selenium [22]. Iodine can also be found in dairy and eggs. Iodine insufficiency is prevalent in Western Europe [20]. In the Netherlands, 60% of iodine intake is derived from non-natural sources. Iodized salt in bread accounts for 50% of the Dutch iodine intake [23]. In 2008, the maximally permitted iodine content of bakery salt in the Netherlands was reduced from 70–85 to 50–65 mg/kg salt. It was simultaneously allowed to add iodine to almost all foods [24]. Since then, iodine intake has decreased by 20–25% [25] or even 30–37% by 2015 [26]. In a pilot of 2018, we showed that pregnant Dutch women have iodine insufficiency [27].

Selenium contents in nuts, meat, grain, vegetables and dairy depend largely on the local soil selenium content and its bioavailability [28]. Europe is a selenium-poor continent, putting its inhabitants at risk of suboptimal selenium status [29]. In the Netherlands, the mean selenium intake of women aged 19–30 and 31–50 years amounts to 33.3 and 47.9 µg/day, respectively, which is well below the Dutch adequate intake of 70 µg/day [30].

There are little data on the combined iodine and selenium status during pregnancy. In a Northern Ireland pregnant cohort, the median urine iodine concentration and mean selenium intake proved insufficient [31], which was also shown in New Zealand [32]. However, the number of women with combined iodine and selenium insufficiency was not reported. Stråvik et al. reported insufficient iodine and selenium intakes and status during pregnancy and lactation in Sweden [33]. Only 5% and 3% of women in their study showed adequate intakes of both iodine and selenium during pregnancy and lactation, respectively. In a cross-sectional study we investigated the iodine status and selenium intake of pregnant women in the northern part of the Netherlands and related the outcome to their supplement use. We measured iodine and selenium concentrations in 24 h urine and calculated iodine excretion and estimated selenium intake. In addition, we evaluated the iodine status by applying the commonly used iodine concentration (UIC) and the iodine/creatinine ratio (UICR) in both 24 h urine and morning urine.

## 2. Materials and Methods

### 2.1. Study Population and Medical Ethical Aspects

This cross-sectional study was mainly conducted in the northern part of The Netherlands. Its primary goal was to investigate the iodine status during pregnancy. The secondary goal was the estimation of selenium intake. Pregnant women were invited to participate in the “JOdium Zwangeren Onderzoek” (JOZO) taking place between February 2018 and February 2019. Invitations occurred via verbal contact and by spreading flyers in a.o. midwife practices and during childbirth classes.

Apparently healthy pregnant women were eligible to participate. Exclusion criteria were multiple pregnancy, (pregnancy-induced) hypertension or diabetes mellitus, renal dysfunction, thyroid disorders and long-term medication use, notably iodine-containing amiodarone and thyroid hormone (thyrax) and also age below 18 years.

All participants received verbal and written information and provided written consent. The Medical Ethical Review Committee of the University Medical Centre Groningen (UMCG) considered this study non-WMO compliant. The study was in agreement with the Declaration of Helsinki of 1975, as revised in 2013.

### 2.2. Data Collection, Storage and Analysis

The participants provided information about their education level, gestational age, number of previous pregnancies and liveborn infants, length, pre-pregnancy weight, use of nutritional supplements, alcohol, drugs, medication and tobacco, and also medical conditions. Women who graduated high school, intermediate vocational education or less were categorized as lowly educated. Women who graduated college, university or higher were highly educated. Using the brand name of the supplement, we could retrieve the daily iodine and selenium doses, if applicable. The available supplements contained no, 75 or 150 µg of iodine and 8.2–60 µg of selenium per daily dose.

Upon instruction, the participants collected the first morning urine portion and subsequently a 24 h urine sample. The samples were stored in a cool place and transported to the UMCG within 48 h. The participants noted the starting and final time of collection. The 24 h urine collections that were incomplete (as noted by the participant), had a volume <300 mL [34] or were not collected within a 22–26 h time frame were considered to be collected incorrectly and therefore excluded. The volume of 24 h urine was estimated by the investigators and subsequently, it was homogenized and divided into portions. All samples were stored at −80 °C. Iodine and selenium concentrations were measured in the Laboratory of Clinical Pharmacy of the UMCG by inductively coupled plasma-mass spectrometry (ICP-MS; Varian-Analytik-Jena Jena Germany.). The intra- and inter-assay coefficients of variation (CVs) were 1.9 and 5.9% at 40 µg/L, and 0.9 and 0.5% at 225 µg/L. Urine creatinine concentrations were analyzed using a Roche Cobas 8000 modular analyzer (Roche, Almere, The Netherlands).

### 2.3. Power Analysis

Based on our pilot study [27], we estimated the minimum number of participants needed to detect a significant difference between high and low iodine status. We anticipated that about 30% of the women would not use iodine-containing supplements, 30%—a supplement containing 75 µg iodine, and 30%—a 150 µg iodine supplement. We used the difference between the iodine status before and after a 150 µg iodine-containing supplement during pregnancy [27] to calculate the minimum number of participants needed with a power of 0.80 and α of 0.05. This resulted in at least 48 subjects in each group (Wilcoxon–Mann–Whitney test). To ensure power, we included 200 participants, since the participants in our pilot study proved highly educated and therefore were not representative of the Dutch pregnant population.

### 2.4. Parameters of Iodine Status and Selenium Intake

The World Health Organization (WHO) advises evaluation of the iodine status of a population by using the iodine concentration of a random spot urine specimen [35]. However, using 24 h urine reduces the fluctuations in iodine excretion during the day and was therefore preferred in our pilot and current studies. The iodine status of a population of pregnant women can be considered sufficient if the median UIC reaches at least 150 µg/L. The WHO assumes a mean urine volume of 1.5 L/day [35]. However, in our pilot study, we found a mean urine volume of 1.9 L/day, indicating that the 24 h UIC is likely to underestimate iodine status. We therefore chose to calculate the 24 h urine iodine excretion (24H-UIE), which was recently described as the gold standard to evaluate iodine nutrition during pregnancy [34], and it was reasoned that an adequate status of the population can be expected when the median 24H-UIE is at least 225 µg (1.5 L* ×150 µg/L). In addition, we used the UICR cut-off value of 150 µg/g to evaluate the iodine status [36,37,38]. Excessive iodine status was defined as a UIC > 500 µg/L [35].

Selenium intake was estimated as (100/55) × 24 h urine selenium excretion. This formula is based on 50–60% excretion of the ingested selenium via urine [39]. The calculated median selenium intake was compared with the nutritional recommendations for pregnant women. The Estimated Average Requirement (EAR) employed by the Institute of Medicine (IOM) amounts to 49 µg/day, giving rise to an RDA of 60 µg/day [40]. The European Food Safety Authority (EFSA) recommends an Adequate intake (AI) of 70 µg selenium/day during pregnancy [41], which is adopted by the Health Council of the Netherlands [42].

### 2.5. Data Analysis

We used IBM-PASW Statistics 23 software for statistical analysis. The relation between gestational age and the investigated iodine status parameters was investigated with Spearman correlation analysis. Between-group differences were analyzed using the Kruskall–Wallis H test with post hoc paired comparisons. The associations between daily iodine dose from supplements and the iodine status parameters were analyzed using the Jonckheere’s Trend test.

## 3. Results

### 3.1. Study Population

A total of 292 apparently healthy pregnant women were willing to participate in the JOZO study. We excluded participants who did not provide written permission, (n = 70), used iodine-containing medication (n = 2) or provided incorrectly collected 24 h urine (n = 19).

Two of the 201 included participants showed unexpectedly high 24 h UIC (987 and 1072 µg/L) and high morning UIC (1752 and 4401 µg/L, respectively). One participant showed a high morning UIC (824 µg/L). Since the exclusion criteria were not applicable, those three were not excluded. Iodine was analyzed in all 201 urine samples, but selenium was merely analyzed in the 24 h urines of 186 participants who provided permission for the measurement of additional analytes.

Table 1 shows the characteristics of the JOZO participants. Data on nutritional supplement use were lacking in 21 subjects. Eighty-six participants (42.8%) reported that they did not use supplements or used supplements without iodine. Iodine-containing supplements were taken by 94 (46.8%) participants. The daily iodine dose in these supplements amounted to 75 (n = 59) or 150 (n = 35) µg. Selenium-containing supplements were used by 100 (49.8%) participants.

### 3.2. Iodine Status

Table 2 shows the iodine status of the JOZO participants. All median iodine status parameters were below their respective cut-off values: 24H-UIE 185 µg (cut-off value 225 µg), 24H-UIC 95 µg/L (cut-off value 150 µg/L), 24H-UICR 141 µg/g (cut-off value 150 µg/g), morning UIC 129 µg/L (cut-off value 150 µg/L) and morning UICR 114 µg/g (cut-off value 150 µg/g). Iodine insufficiency was found in 69.7% of the participants using the 24H-UIE and varied from 57.7% (24H-UICR) to 88.1% (24H-UIC). The median 24 h urine volume of 1.9 L was higher than the 1.5 L assumed by the WHO. Morning urine (median creatinine concentration 1.07 g/L) seemed more concentrated than 24 h urine (0.67 g/L). There was no relationship between gestational age and the investigated iodine status parameters.

### 3.3. Association between Iodine Supplement Dose and Iodine Status Parameters

Figure 1 shows the relationship between the iodine supplement dose and the 24 h UIE (panel A; in µg) and also the 24 h-UIC (panel B; in µg/L). The Kruskal–Wallis H test revealed between-dosage-group differences in iodine excretion and urine iodine concentration (*p* < 0.011). The Jonckheere’s Trend test showed a positive association of iodine dose with 24 h iodine excretion (*p* = 0.000), UIC (*p* = 0.002) and UICR (*p* = 0.000), and with morning UIC (*p* = 0.003) and UICR (*p* = 0.000).

The 24 h iodine excretion of the participants who did not use iodine-containing supplements (median; range: 157; 66–1629 µg; Figure 1A) was significantly lower than the iodine excretion of their counterparts using 75 µg iodine supplements (211; 60–329 µg; *p* = 0.04) and counterparts using 150 µg iodine supplements (244; 43–2654 µg; *p* = 0.02). Accordingly, the 24 h UIC of participants who did not use iodine-containing supplements (91; 31–1072 µg/L; Figure 1B) was significantly lower than the 24 h UIC of counterparts using 150 µg iodine supplements (114; 53–987 µg/L; *p* = 0.02).

Women who used a 150 µg iodine supplement showed significantly higher 24H-UICR (171; 108–1938 µg/g) than did counterparts using 75 µg iodine supplements (148; 61–275 µg/g; *p* = 0.01) and those who did not use iodine-containing supplements (130; 56–1200 µg/g; *p* < 0.001). The 24H-UICR of participants who took 75 µg iodine was significantly higher than the 24H-UICR of those who did not take iodine-containing supplements (*p* = 0.031).

The morning UIC (156; 75–1752 µg/L) and morning UICR (143; 70–1702 µg/g) of women who took a 150 µg iodine supplement were higher than those of counterparts who did not use those supplements (UIC 115; 34–4401 µg/L; *p* = 0.008 and UICR 101; 42–6282 µg/g; *p* = 0.000).

### 3.4. Iodine Supplement Dose That was Associated with Sufficient Iodine Status

The median iodine status parameters of the participants who did not take supplements containing iodine amounted to: 24 h iodine excretion, 157 µg (cut-off value 225 µg); 24 h UIC, 91 µg/L (cut-off value 150 µg/L); 24 h UICR, 130 µg/g (cut-off value 150 µg/g); morning UIC, 115 µg/L (cut-off value 150 µg/L); and morning UICR, 101 µg/g (cut-off value 150 µg/g). Based on all five parameters, the iodine status of those participants proved insufficient.

The iodine status of participants who took a 75 µg iodine supplement proved insufficient as well (medians: 211 µg, 94 µg/L, 148 µg/g, 131 µg/L and 119 µg/g, respectively).

For participants who took 150 µg iodine supplements, we found the following: median 24 h iodine excretion, 244 µg; 24 h UIC, 114 µg/L; 24 h UICR, 171 µg/g; morning UIC, 156 µg/L; and morning UICR, 143 µg/g. Their iodine status proved sufficient when based on 24 h iodine excretion, 24 h UICR and morning UIC but not when based on 24 h UIC and morning UICR.

### 3.5. Selenium Intake in Relation to Nutritional Recommendations

The median estimated selenium intake amounted to 48 µg/day (range: 12-180). Of the 186 participants, 53.8% showed a estimated selenium intake below the 49 µg/day IOM-EAR, 77.4% a estimated selenium intake below the 60 µg/day RDA (IOM, Nordic) and 88.7% an intake below the 70 µg/day AI (Health Council of the Netherlands, EFSA). The 78 participants who did not take selenium-containing supplements showed a median calculated intake of 42 (16–90) µg/day. Their intakes did not reach the 49 µg/day EAR in 71.8% and did not reach the 60 µg/day RDA in 97.4%; two subjects (2.6%) had selenium intakes above the 70 µg/day AI. Figure 2 shows the estimated selenium intake in relation to the reported daily doses of selenium in nutritional supplements. There was no clear relationship between selenium supplemental dose and estimated selenium intake. Consequently, we were unable to establish the supplemental selenium dose that provides an adequate intake.

### 3.6. Combined Insufficient Iodine Status and Inadequate Selenium Intake

Figure 3 shows 24 h urine iodine excretion versus the estimated selenium intake for the 186 JOZO participants who allowed us to measure both iodine and selenium. It shows that 68.3% of them had insufficient iodine status (WHO: 225 µg), while 77.4% showed selenium intakes below the 60 µg/day RDA (IOM, Nordic). The percentages of combined insufficient iodine status and insufficient selenium intakes were 46.8% below the 49 µg/day EAR, 61.3% below the 60 µg /day RDA and 64.5% below the 70 µg /day AI.

## 4. Discussion

The iodine status of pregnant Dutch women proved inadequate: the medians of all applied iodine status parameters were below their respective cut-off values. Only the use of a supplement containing 150 µg iodine was associated with adequate iodine status during pregnancy. A new finding was that the median estimated selenium intake (48 µg/day) was below the 49 µg/day EAR, that 77.4% of the pregnant women consumed less than the 60 µg selenium RDA and that 88.7% consumed less than the 70 µg/day AI. The selenium intakes of the 78 participants who did not take selenium-containing supplements did not reach the 60 µg RDA in 97.4%. The estimated selenium intake proved independent of the selenium supplemental dose, which made it impossible to estimate how much supplemental selenium is needed to reach an adequate intake. Dependent on the applied cut-offs, 46.8–64.5% of all participants had a combined inadequate iodine status and insufficient selenium intake.

The 150 µg/L (UIC) and 150 µg/g (ICR) cut-offs for iodine in random urine portions, as advised by the WHO, are commonly used to establish a population’s iodine status. Although the UIC (median 129; 334–4401) proved higher in morning urine than in the 24 h collections, (95; 31–1072), the morning urine appeared more concentrated, as suggested by its higher creatinine concentration. Consequently, the ICR (114; 37–6282 µg/g) in the morning urine was lower than the ICR in the 24 h collections (141; 42–1938 µg/g). These data show that the interpretation of the ICR and UIC of random urine samples is difficult. A small study with 20 apparently healthy individuals by us showed that the mean intraindividual coefficients of variation of the UIC and ICR, measured in consecutively collected urine portions during 24 h, amounted to 52 and 26%, respectively (unpublished data). The 24 h urine collections are obviously more representative than random portions [43,44,45,46]. Both this study and our pilot study show that the use of a 24 h UIC underestimates the iodine status, because the encountered median urine volumes (1.9–2.0 L) were higher than the 1.5 L used by the WHO [35]. We therefore conclude that during pregnancy, 24 h urine iodine excretion is the most reliable parameter of iodine status, as recently demonstrated [34].

The median 24 h urine iodine excretion of the whole group (185 µg) and of women taking a 150 µg supplement (244 µg) in the current JOZO study were lower than those in our pilot study (week 20: 210 µg; week 36: 304 µg) [27]. The current median 24 h UIC (95 µg/L) and morning UIC (129 µg/L) were low but in harmony with the medians of 87, 88, 101, 114 and 124 µg/L found in random urine samples of pregnant women living in Belgium [47], the United Kingdom [48], Sweden [49,50], Denmark [51] and Austria [52], respectively. Both the 24 h and morning UIC, as well as the 24 h (141 µg/g) and morning (114 µg/g) UICR of the JOZO participants were lower than the median UIC (229.6 µg/L) and ICR (296.5 µg/g) of pregnant women in the Dutch Generation-R study [38]. The latter study was conducted prior to the lowering of the maximum iodine content of bakery salt in 2008. Therefore, the current study confirms the decline of the Dutch iodine status since 2008, as previously established by others [25,26].

The estimated selenium intake (whole group 48 µg/day (12–180); non-supplement users 42 (16–90) µg/day) of the current 31 (21–43)-year-old pregnant Dutch women proved to be in agreement with the selenium intakes from food and dietary supplements reported by the Dutch National Food Consumption Survey 2012–2016 for 4313 19–79-year-old subjects (P50 (P5-P95): men 55 (33–101); women 44 (24–95) ug/day)) [53]. Since pregnant women should comply to higher selenium intakes than non-pregnant counterparts, this implies that the currently investigated women are even more selenium deficient than the general Dutch population. The various formulations of selenium in commercially available supplements may have introduced differences in bioavailability which may explain the failure to find a relationship between supplement dose and estimated selenium intake. For instance, selenomethionine is readily retained in the body as compared to selenite [54].

Although selenium excretion at the same supplement dose was higher in pregnant women as compared to that of non-pregnant counterparts [54], a more recent study showed that the median selenium intake estimated from three 24 H dietary recalls (51 ug/day) was only marginally higher than that based on 24 h urine excretion (49 ug/day) in pregnant women [32]. We therefore conclude that the calculation of selenium intake from 24H urine selenium excretion is also reliable in pregnancy.

Intervention studies do not unambiguously indicate that iodine supplementation during pregnancy improves offspring neurological development [44], which can be explained by differences in the timing of the supplement. Early iodine supplementation proves most effective in preventing fetal neurological dysfunction [44]. Iodine is needed from early pregnancy to support increased production of maternal thyroid hormone [55]. The WHO and European Thyroid Association (ETA) recommend a daily total iodine intake of at least 250 µg, using supplements with 150–250 µg iodine, preferably starting before conception [35,56]. Iodine supplementation during pregnancy may, in women with mild iodine insufficiency, increase serum TSH (the Wolff–Chaikoff effect), which is explained by thyroid auto regulation that follows a sudden iodine influx [57]. However, a more recent study showed that a 200 µg iodine/day supplement during pregnancy caused no difference in TSH, slightly higher fT4 and T4 and lower thyroglobulin [58].

Negro et al. [59] showed that a 200 µg selenomethionine supplement during pregnancy and in the postpartum period increased the serum selenium concentration, prevented its decrease during pregnancy and also reduced thyroid inflammatory activity and the incidence of hypothyroidism. However, an inverse U-shaped relationship between maternal selenium status and offspring neurological development was found in Spain [60]. The relationship suggests that during pregnancy, a maternal selenium concentration of 80 µg/L is optimal [60]. Little is known about the effects of combined selenium and iodine insufficiency. Data from African countries suggest that iodine deficiency combined with selenium sufficiency protects the thyroid against peroxidative damage induced by augmented TSH, although it coincides with neurological cretinism. When both iodine and selenium are deficient, peroxidative damage to thyrocytes was found, whereas neurological damage was limited [61]. Based on these scarce data, simultaneous supplementation of iodine and selenium seems advisable.

There are a number of limitations in this study. The studied pregnant women were highly educated. This happened notwithstanding our attempts to recruit a representative sample by including more women than statistically needed. Highly educated women are more likely to use nutritional supplements and usually eat a healthier diet. A positive association between education level and iodine intake was previously found in Dutch adults [62], and education also related with iodine status in pregnant Hungarian women [63]. This suggests that the iodine status of a representative group might be even worse than the one currently reported. In addition, we did not analyze the iodine and selenium contents of the supplements ourselves and cannot be sure that supplements were taken on a daily basis. Although food frequency questionnaires are subject to under- and over-reporting of food intake, the combination of an FFQ and urinary selenium excretion may have been more reliable to estimate selenium intake. Urine samples were collected only once, which according to the WHO is sufficient for population estimates but not for estimating the iodine status of individuals. We were not able to evaluate the completeness or correctness of urine collection (e.g., at low and high volumes) using creatinine excretion, since reference values for pregnant women were lacking. In addition, there were no restrictions in meat consumption and exercise, which may have influenced urine creatinine concentrations.

## 5. Conclusions and Recommendations

The majority of pregnant Dutch women in our study had inadequate status and an insufficient selenium intake. According to well-established associations in the literature, the mothers may be considered at risk of thyroid disfunction, including thyroid autoimmune disease and cancer, while their offspring are at risk of neurological development and possibly suboptimal growth [64] The poor iodine status seems at least partially caused by the lowering of bakery salt iodine content effective since 2008. A logical measure would be to return to the pre-2008 iodine contents. However, Dutch women in childbearing age nowadays show low intakes of bread and grains. Only 31.5% and 36.2% of 19–30 years and 31–50 years old Dutch women, respectively, meet the recommendations for bread and grains [65].

It has become clear that iodine and selenium interact and that many diseases linked to either iodine or selenium in the past are in reality caused by an iodine–selenium disbalance [17]. As seafood is a rich source of both iodine and selenium, it would be wise to revise the current Dutch recommendation for the consumption of fish and of seafood in general. These recommendations were in 2015 revised from “eat fish 2 times a week, with one portion from fatty fish” to “one time fish, preferably fatty fish” [66]. Consequently, together with Bosnia and Herzegovina, The Netherlands nowadays has the lowest recommendation for fish intake within Europe, translating to 100 g of fish a week [67]. At present, only a minority of women of childbearing age (19% and 24.3% of 19–30 years and 31–50 years old women, respectively) meets this recommendation [65]. Dietary iodine and selenium are currently derived at only 4.0 and 9.4% from fish [53]. Returning to the 2015 recommendation, and, notably, an improvement of compliance, would not only improve iodine and selenium status but also that of other nutrients that together with iodine and selenium have been coined “brain selective nutrients”, notably vitamins A, D and B12, iron, zinc, copper and fish oil fatty acids eicosapentaenoic acid and docosahexaenoic acid [68]. Such a measure is not in line with sustainability but gives credit to the fact that humans eat food, not nutrients. Global warming, the emptying of the sea and other contemporary worldwide challenges have not changed our basic physiology. An alternative is to promote the consumption of seaweed, which is a rich source of both iodine and selenium because of their bioconcentration from seawater. They also bioconcentrate toxic heavy metals [22], but the risk of heavy metals in food with concomitantly high selenium contents is exaggerated and applies notably to some (older) top-level predatory fish [69,70,71].

It may be wise to educate women who plan pregnancy that iodine needs increase already in early pregnancy and to recommend augmentation of their iodine consumption, preferably from seafood and/or seaweed, prior to conception. This can also be achieved by taking a 150 µg iodine/day supplement, which is in line with the advice of the WHO and ETA [35,56]. The results of this study indicate that selenium should be added to this advice, although the optimal dose and formulation need additional investigation. However, the advice to take a supplement during pregnancy disregards the probably equally poor iodine and selenium status of the nonpregnant Dutch population and the interaction of iodine and selenium in the etiology of many diseases [64].

## Figures and Tables

**Figure 1 nutrients-14-03936-f001:**
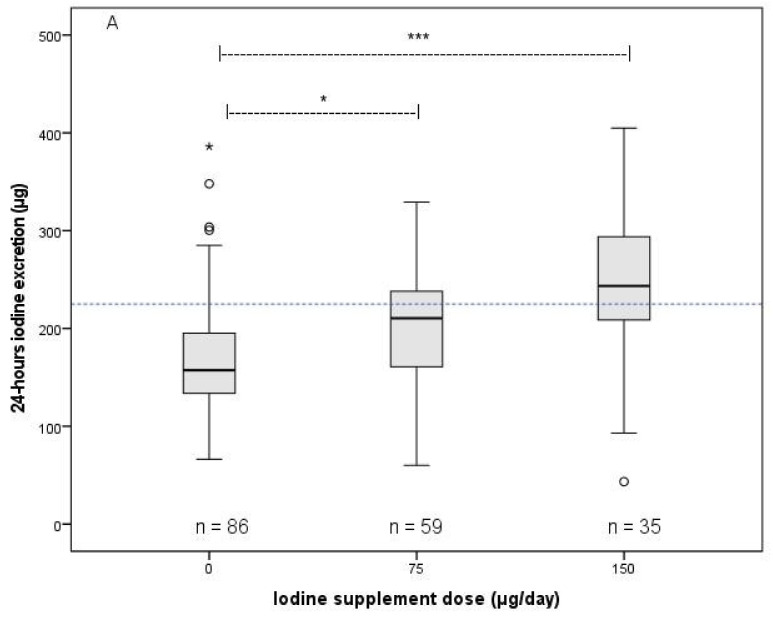
Iodine excretion in 24 h urine (panel (**A**); in µg) and 24 h UIC (panel (**B**); in µg/L) of pregnant JOZO participants in relation to the reported iodine supplement dose. Cut-off values in panel A: 225 µg and panel B: 150 µg/L. These cut-off values are used for populations, not individuals. * *p* < 0.05, *** *p* ≤ 0.001. Dose 0: pregnant participants who did not use nutritional supplements (n = 32) + counterparts who used supplements but without iodine (n = 54). For clarity, two subjects with excessive iodine status (see text) were not included in the figures. Abbreviation: UIC, urine iodine concentration.

**Figure 2 nutrients-14-03936-f002:**
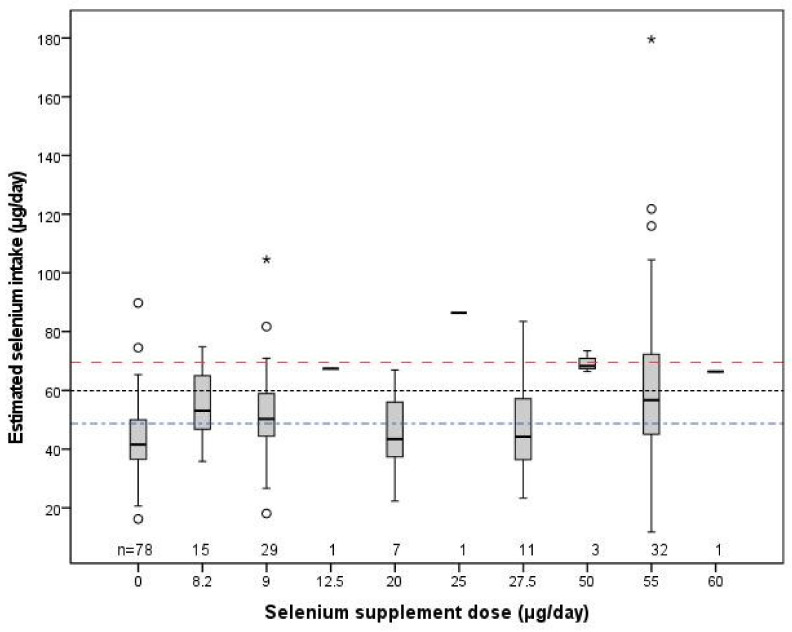
Estimated selenium intake (µg/day) of pregnant JOZO participants in relation to the reported selenium supplement dose (µg/day). The lower horizontal line (-∙-∙) depicts the 49 µg/day Estimated Average Requirement (EAR) of the IOM, the middle horizontal (∙∙∙∙) line—the 60 µg/day RDA of the IOM and Nordic, and the upper horizontal line (----) shows the 70 µg/day Adequate Intake (AI) of the EFSA, as adopted by the Health Council of the Netherlands.

**Figure 3 nutrients-14-03936-f003:**
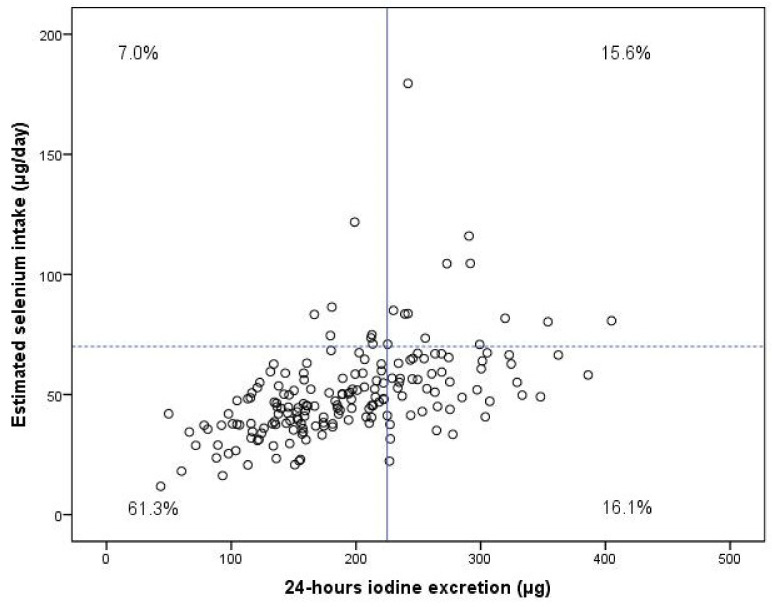
Estimated selenium intake (µg/day) in relation to 24 h urine iodine excretion (µg). The horizontal line (∙∙∙∙) depicts the 60 µg/day RDA (IOM, Nordic) for pregnant women, and the vertical line (|) depicts the 225 µg cut-off value for 24 h urine iodine excretion.

**Table 1 nutrients-14-03936-t001:** Characteristics of the pregnant women included in the JOZO study.

Variable	Unit	Median (Range)/n (%)
Age	years	31 (21–43)
Pre-pregnancy BMI	kg/m^2^	23 (17–51)
Gestational age	weeks	29 (6–42)
Number of pregnancies	n	2 (1–6)
Parity	n	1 (0–3)
Education		
High	n (%)	158 (78.6)
Low	n (%)	43 (21.4)
Nutritional supplements		
Unknown	n (%)	21 (10.4)
No supplement	n (%)	32 (15.9)
Use of supplement	n (%)	148 (73.6)
Supplement without iodine	n (%)	54 (26.9)
Iodine-containing supplement	n (%)	94 (46.8)
75 µg/day	n (%)	59 (29.4)
150 µg/day	n (%)	35 (17.4)
Selenium-containing supplement	n (%)	100 (49.8)

Data are depicted as median (range) or number (n) and percentage of the 201 participants (n (%)). One pregnant woman consumed alcohol, and 5 reported tobacco use. Low education: women who graduated high school, intermediate vocational education or lower; high education: women who graduated college, university or higher. BMI is body mass index.

**Table 2 nutrients-14-03936-t002:** Iodine status parameters and urine creatinine concentrations of the pregnant JOZO participants.

Parameter	Unit	Median	Range			% Insufficient	Cut-Off Value
24 h UIE	µg	185	43	-	2654	69.7	225
24 h UIC	µg/L	95	31	-	1072	88.1	150
24 h ICR	µg/g	141	42	-	1938	57.7	150
Morning UIC	µg/L	129	34	-	4401	64.3	150
Morning ICR	µg/g	114	37	-	6282	73.4	150
24 h urine volume	mL	1940	400	-	5550		
Creatinine concentration:							
24 h urine	g/L	0.67	0.21	-	2.73		
Morning urine	g/L	1.07	0.21	-	4.09		

Iodine status is expressed as: 24 h urine iodine excretion, 24 h UIC, 24 h ICR, Morning UIC and morning ICR. Abbreviations: UIE, urine iodine excretion; UIC, urine iodine concentration; ICR, iodine/creatinine ratio.

## Data Availability

No publicly archived datasets were analyzed or generated during the study. The data presented in this study are available on request from the corresponding author.

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
