# Peer review of "Pregnant Dutch Women Have Inadequate Iodine Status and Selenium Intake"

_nutrients, 2022, doi:10.3390/nu14193936_

Round 1

Reviewer 1 Report

1.     Overall description of the study:

The manuscript reports a cross-sectional descriptive study on the iodine status and selenium intake of pregnant women living in northern part of The Netherlands. Subjects are participant of JOZO study. Authors assessed the iodine status and compared the results obtained by 24H-UIE, 24H-UIC, 24H-ICR, morning UIC and morning ICR. Selenium intake was estimated by 24H urinary selenium excretion.

2.     Comments and suggestions to authors

(1)   Since The Netherlands locates in the European continent with poor soil selenium content, and has experienced policy change regarding iodine content of bakery salt, this descriptive cross-sectional study provides novel information that geographic characteristics and policy decision may influence the nutritional status of critical micronutrients for pregnant women and their fetus. Readers may gain information and knowledge through the results of the problems encountered and the solutions taken by the country or authors reported in the manuscript.

(2)   One of the major findings reported by authors is using 24H urinary iodine excretion as the key parameter for accessing iodine status of pregnant women. For this reason, it is critical to make sure that the 24H urine and its volume were correctly or properly collected and recorded. Authors are suggested to describe the related process in section of Data collection, storage, and analysis (Line 115-120). Was 24H urinary creatinine excretion matched with the lean body mass or height of subjects or within a reasonable range?

(3)   Since the range of 24-hour urine volume reported int Table 2 is as low as 400 ml and as high as 5500 ml, please explain if these extreme values are reasonable daily total urine volume for the target population of this study (apparently healthy pregnant women, Line 98).

(4)   Estimating selenium (Se) intake by 24H urinary selenium excretion is also a special finding of this study. Dietary selenium intake usually is measured from the Se content of collected meals or calculated from dietary assessment tools with the aid of Se contents in food items from nutrient data base. Could authors provide dietary Se intakes through these ways?

(5)   Authors estimated dietary Se intake based on 24H urinary Se excretion. The formula used is based on a 50-60% excretion rate reported in reference 36 (Line 151). The data of Reference 36 came from the study on 62 adults (31 men, 31 women). The subjects of this manuscript are pregnant women. Please comment on whether the metabolic rate or Se retention rate of pregnant women are same as those of healthy adults. Since the absorption rate or metabolism of many nutrients changed (a lot of time enhanced) during pregnancy, estimating dietary intake of nutrients based on urinary excretion must be interpreted with caution. Authors need to verify the estimation made in this study would not overestimate or underestimate the dietary Se intake.

(6)   Since Se supplement use is not related with the Se intake estimated from urinary Se excretion, authors are suggested to comment the reason behind this observation. One reason might be the soundness of Se intake estimation in this study; another reason might be the forms of Se supplements used by subjects. If the Se came from Se yeast, its Se is usually in the form of selenomethionine, it may be absorbed and incorporated into body protein instead of excreted in urine.

(7)   There are many issues which may require caution in result interpretation. For example, urine samples were collected in only 1day, Se intake was indirectly estimated from urinary excretion instead of measured directly, whether subjected were restricted from strenuous exercise or large amount of meat consumption to avoid enhanced urinary creatinine excretion, etc. Authors may consider these issues and mention them in Limitations.

(8)   Panels A and B are too small to read.

(9)   For Figure 2, authors may consider setting the upper limit of Y axis to 150 (ug/day) since it seems no data are above 150. And the lower part of the figure 2 may be scaled up (enlarged) to be read easier.

(10)      The format of “References” needs to be carefully revised to fit the requirements of the journal. The language of many non-English references need to be indicated.

Author Response

Reviewer 1:

Comments and suggestions to authors

(1)   Since The Netherlands locates in the European continent with poor soil selenium content, and has experienced policy change regarding iodine content of bakery salt, this descriptive cross-sectional study provides novel information that geographic characteristics and policy decision may influence the nutritional status of critical micronutrients for pregnant women and their fetus. Readers may gain information and knowledge through the results of the problems encountered and the solutions taken by the country or authors reported in the manuscript.

(2)   One of the major findings reported by authors is using 24H urinary iodine excretion as the key parameter for accessing iodine status of pregnant women. For this reason, it is critical to make sure that the 24H urine and its volume were correctly or properly collected and recorded. Authors are suggested to describe the related process in section of Data collection, storage, and analysis (Line 115-120). Was 24H urinary creatinine excretion matched with the lean body mass or height of subjects or within a reasonable range?

We thank the reviewer for this valuable comment. We described in the Materials and methods section, that collections that were incomplete or were not collected within a 22-26 hours timeframe, were exclude. We added the volume < 300 ml as used by Bu during pregnancy. To be clear that the urine volume was not estimated by the participants, we added that it was established by one of the investigators.

  Lines 124-127: “The 24-hour urine collections that were incomplete (as noted by the participant), had a volume < 300 ml [34] or were not collected within a 22-26 hours time frame were considered to be collected incorrectly and therefore excluded. The volume of the 24-hour urine was estimated by the investigators”

 We were not able to use appropriate reference values for creatinine excretion during pregnancy as they are currently lacking for Dutch pregnant women and using the same or comparable techniques. We added this to the limitations.

Lines 391-393: “ We were not able to evaluate the completeness or correctness of urine collection (e.g. at low and high volumes) using the creatinine excretion, since reference values for pregnant women were lacking.”

(3)   Since the range of 24-hour urine volume reported int Table 2 is as low as 400 ml and as high as 5500 ml, please explain if these extreme values are reasonable daily total urine volume for the target population of this study (apparently healthy pregnant women, Line 98). We used the cut-off value of 300 mL as stated above to ensure that the urine collection was complete.

The high volume may indicate incorrect urine collection, and most likely exceeding the 26 hr timeframe. A 24H urine volume >5 L is not unusual in our lab, and therefore was not commented. We added to the limitations that we were not able to evaluate the correctness and completeness of the urine collection, e.g. at low and high volumes, using creatinine excretion. See answer to comment 2.

(4)   Estimating selenium (Se) intake by 24H urinary selenium excretion is also a special finding of this study. Dietary selenium intake usually is measured from the Se content of collected meals or calculated from dietary assessment tools with the aid of Se contents in food items from nutrient data base. Could authors provide dietary Se intakes through these ways?

Unfortunately, not. Although we collected data on nutritional intake, we did not use an FFQ that was suitable for this purpose. It should however be noted that also FFQ are subject to over- and underreporting of food.

(5)   Authors estimated dietary Se intake based on 24H urinary Se excretion. The formula used is based on a 50-60% excretion rate reported in reference 36 (Line 151). The data of Reference 36 came from the study on 62 adults (31 men, 31 women). The subjects of this manuscript are pregnant women. Please comment on whether the metabolic rate or Se retention rate of pregnant women are same as those of healthy adults. Since the absorption rate or metabolism of many nutrients changed (a lot of time enhanced) during pregnancy, estimating dietary intake of nutrients based on urinary excretion must be interpreted with caution. Authors need to verify the estimation made in this study would not overestimate or underestimate the dietary Se intake.

Thank you for this valuable comment. Indeed, based upon a supplementation study, it was estimated that at the same dose, the urinary excretion of selenium was higher during pregnancy. However, in a more recent study the comparison of median selenium intake based on 24H recalls and 24H urine excretion showed only marginal differences.

We added to the discussion: lines 347-352:

“Although selenium excretion at the same supplement dose was higher in pregnant` women as compared to that of non-pregnant counterparts [54], a more recent study showed that the median selenium intake estimated from three 24 H dietary recalls (51 ug/day) was only marginally higher than that based on 24-H urine excretion (49 ug/day) in pregnant women [32]. We therefore conclude that the calculation of selenium intake from 24H urine selenium excretion is also reliable in pregnancy.”

(6)   Since Se supplement use is not related with the Se intake estimated from urinary Se excretion, authors are suggested to comment the reason behind this observation. One reason might be the soundness of Se intake estimation in this study; another reason might be the forms of Se supplements used by subjects. If the Se came from Se yeast, its Se is usually in the form of selenomethionine, it may be absorbed and incorporated into body protein instead of excreted in urine.

Correct, this was stated in the limitations, and we decided to add this to the discussion and give more information.

Lines 343-346: “ The various formulations of selenium in commercially available supplements may have introduced differences in bioavailability which may explain the failure to find a relation between supplement dose and estimated selenium intake. For instance, selenomethionine is readily retained in the body as compared to selenite [54].“

(7)   There are many issues which may require caution in result interpretation. For example, urine samples were collected in only 1day, Se intake was indirectly estimated from urinary excretion instead of measured directly, whether subjected were restricted from strenuous exercise or large amount of meat consumption to avoid enhanced urinary creatinine excretion, etc. Authors may consider these issues and mention them in Limitations.

The estimation of selenium intake using the seleniun excretion was discussed as stated above.

We added to the limitations, lines 387-395:

Although food frequency questionnaires are subject to under and overreporting of food intake, the combination of an FFQ and the urinary selenium excretion may have been more reliable to estimate selenium intake. Urine samples were collected only once, which according to the WHO is sufficient for population estimates but not for estimating the iodine status of individuals. We were not able to evaluate the completeness or correctness of urine collection (e.g. at low and high volumes) using the creatinine excretion, since reference values for pregnant women were lacking. In addition, there were no restrictions in meat consumption and exercise, which may have influenced urine creatinine concentrations.

(8)   Panels A and B are too small to read.

Agree and adjusted. We added higher quality figures to the manuscript.

We will ask the editorial office to judge the figure quality.

(9)   For Figure 2, authors may consider setting the upper limit of Y axis to 150 (ug/day) since it seems no data are above 150. And the lower part of the figure 2 may be scaled up (enlarged) to be read easier.

Figure 2 Y axis adjusted

(10)      The format of “References” needs to be carefully revised to fit the requirements of the journal. The language of many non-English references need to be indicated.

Thank you, we adapted accordingly. (see references)

Reviewer 2 Report

The study explores the iodine status and selenium intake in relation to iodine and selenium supplement use in pregnant Dutch women.

The scope of the work is well explained and supported previous studies. The materials and methods, as well as the results are clearly reported in the text, less in the graphic part that needs improvement. The article can be accepted by improving some parts:

Line 83: specify at least one of the studies to which you refer.

Figure 1 is very small and difficult to interpret the results it illustrates. Please correct them.

Remove line 357 and insert this short part into the discussion.

Line 374. It is not clear which bibliographic reference it refers to, alternatively add another reference. 

Author Response

Reviewer 2

We thank the reviewer for the very useful comments, that helped us to improve the manuscript.

Line 83: specify at least one of the studies to which you refer. We added the results of 3 studies in pregnancy and rephrased as follows (lines 84-90)

" In a Northern Ireland pregnant cohort, the median urine iodine concentration and mean selenium intake proved insufficient [31], which was also shown in New Zealand [32]. However, the number of women with combined iodine and selenium insufficiency was not reported. Stråvik et al reported insufficient iodine and selenium intakes and status during pregnancy and lactation in Sweden [33]. Only 5% and 3% of women in their study showed adequate intakes of both iodine and selenium during pregnancy and lactation respectively."

Figure 1 is very small and difficult to interpret the results it illustrates. Please correct them.

Added better visible Figures to the manuscript and hope the uploaded figures are of sufficient quality. For this, we ask the editorial office whether the quality is sufficient.

Remove line 357 and insert this short part into the discussion.

This line is removed. Limitations are added

Line 374. It is not clear which bibliographic reference it refers to, alternatively add another reference. The first author name and name of review are added (lines 401-403)

Dijck-Brouwer et al, Thyroidal and extra-thyroidal requirements for iodine and selenium, A combined evolutionary and (patho)physiological approach, accepted for publication in Nutrients this issue].
